# Inference of captions from histopathological patches

**Masayuki Tsuneki**[1]                                                    TSUNEKI@MEDMAIN.COM

**Fahdi Kanavati**[1]                                                    FKANAVATI@MEDMAIN.COM

[1]*Medmain Research, Medmain Inc., 2-4-5-104, Akasaka, Chuo-ku, Fukuoka, 810-0042 Japan*

## Abstract

Computational histopathology has made significant strides in the past few years, slowly getting closer to clinical adoption. One area of benefit would be the automatic generation of diagnostic reports from H&E-stained whole slide images which would further increase the efficiency of the pathologists' routine diagnostic workflows. In this study, we compiled a dataset (PatchGastricADC22) of histopathological captions of stomach adenocarcinoma endoscopic biopsy specimens, which we extracted from diagnostic reports and paired with patches extracted from the associated whole slide images. The dataset contains a variety of gastric adenocarcinoma subtypes. We trained a baseline attention-based model to predict the captions from features extracted from the patches and obtained promising results. We make the captioned dataset of 262K patches publicly available.

**Keywords:** Histopathology, caption prediction, adenocarcinoma, stomach

## 1. Introduction

Deep learning has enabled a large number of applications and advances in computational histopathology encompassing tasks such as classification, cell segmentation, and outcome prediction (Hou et al., 2016; Madabhushi and Lee, 2016; Litjens et al., 2016; Kraus et al., 2016; Korbar et al., 2017; Luo et al., 2017; Coudray et al., 2018; Wei et al., 2019; Gertych et al., 2019; Bejnordi et al., 2017; Saltz et al., 2018; Campanella et al., 2019); and it is slowly getting closer to adoption in clinical workflows. While the detection and classification of cancer is of high importance, pathologists typically write an associated diagnostic report based on their findings from viewing Hematoxylin and Eosin (H&E) stained slides. The automatic generation of diagnostic reports could make the prediction outputs from models more amenable to interpretation, and provide the pathologists more information in their decision making process.

Over the past few years there has been rapid progress in the development of vision language models with architectures based on Recurrent Neural Networks (RNNs) or more recent transformers (Mao et al., 2014; Xu et al., 2015; Li et al., 2019; Huang et al., 2019; Cornia et al., 2020; Wang et al., 2021). See Stefanini et al. (2021) for a review on image captioning. The typical architecture with RNN-based models involves a CNN for the feature extraction and an RNN as a decoder for generating the text.

In the medical context, the generation of reports in radiology have been investigated by several works (Shin et al., 2016; Jing et al., 2017). In particular, for histopathology,

Zhang et al. (2017) aimed at generating structured pathology reports based on a bladder cancer image dataset consisting of only 1,000 500x500 patches extracted from a cohort of 32 patients. More recently, Gamper and Rajpoot (2021) aimed at obtaining general image features from pre-training on image captions using histopathology images extracted from textbooks and compiled in the ARCH dataset; however, the ARCH dataset consists of images extracted from textbooks, which are of mixed quality, magnifications, and resolutions. In our case, we aimed to specifically create a curated dataset of gastric adenocarcinoma cases of consistent quality and resolution extracted directly from WSIs.

In this paper, we aimed at compiling a large dataset (PatchGastricADC22) consisting of 262,777 patches extracted from 991 Whole Slide Images (WSI) of H&E-stained gastric adenocarcinoma specimens with associated diagnostic captions extracted directly from existing medical reports. This roughly approximates the real-world setting where a single diagnostic report is associated with a large WSI. In addition, we trained a few baseline attention models for the prediction of captions from the associated patches. Dataset and code available at https://zenodo.org/record/6021442 and https://github.com/masatsuneki/histopathology-image-caption, respectively.

## 2. Dataset

We obtained a dataset of 991 H&E-stained slides from distinct patients from the surgical pathology files of International University of Health and Welfare, Mita Hospital (Tokyo, Japan). The slides were digitised into WSIs at a magnification of x20. All of collected cases were diagnosed as having adenocarcinoma and were reviewed by three pathologists to confirm the diagnoses. We randomly selected the cases so as to reflect more closely the clinical distribution of adenocarcinoma subtypes. We then extracted the histopathological captions from their corresponding diagnostic reports. The reports were translated from Japanese into English by two expert pathologists. The vocabulary consisted of 277 words (see word cloud in Fig. B5) with a maximum sentence length of 50 words. All the cases were obtained from a single hospital, and, therefore, some of the cases with similar diagnoses had identical text descriptions with little variety – this was also reflected in the translation. As the dataset collection was retrospective, the original reports were generated by different pathologists at the hospital that were not involved in this study.

We then extracted 300x300px patches mostly from regions containing adenocarcinoma lesions at two different magnifications x10 and x20. This was done by initially loosely annotating the specimens only on the adenocarcinoma regions to minimise the extraction of non-adenocaricoma regions. The specimens, as well as the adenocarcinoma regions within the specimens, were of variable sizes; this resulted in a variable number of patches from each WSI.

This resulted in a variable number of patches associated with a given caption. At a magnification of x20, this resulted in 262,777 tiles and at x10, 67,125 tiles. Figure B6 provides the distribution of the number of patches per WSI. Figure 1 shows examples of tiles extracted from WSIs and their corresponding text captions. We divided the dataset randomly into 70% training, 10% validation, and 20% test, stratifying by the subtype. Table 1 provides a breakdown of the adenocarcinoma subtypes in the dataset.

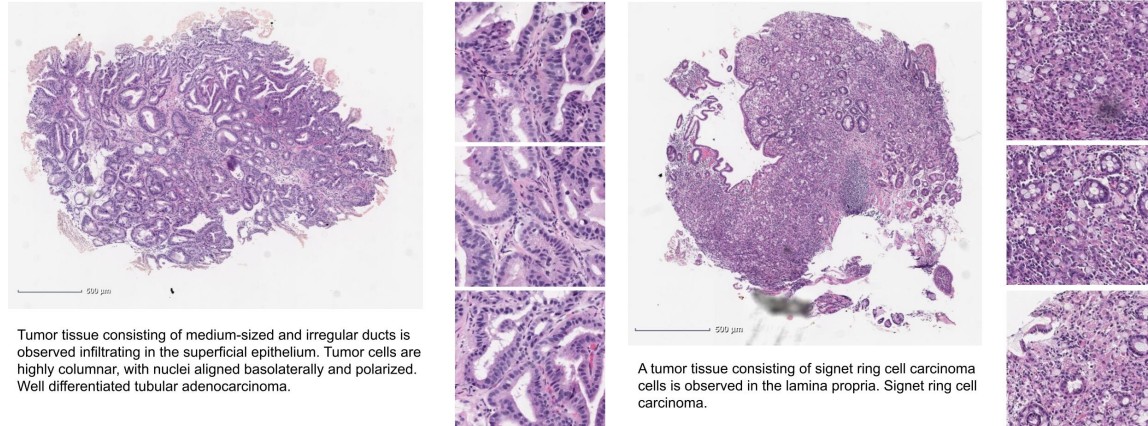

Tumor tissue consisting of medium-sized and irregular ducts is observed infiltrating in the superficial epithelium. Tumor cells are highly columnar, with nuclei aligned basolaterally and polarized. Well differentiated tubular adenocarcinoma.

A tumor tissue consisting of signet ring cell carcinoma cells is observed in the lamina propria. Signet ring cell carcinoma.

Figure 1: Examples of biopsy specimens with associated captions and 300x300px tiles extracted at x20 magnification.

| subtype | count |
|---|---|
| Well differentiated tubular adenocarcinoma | 283 |
| Moderately differentiated tubular adenocarcinoma | 265 |
| Papillary adenocarcinoma | 135 |
| Moderately to poorly differentiated adenocarcinoma | 81 |
| Poorly differentiated adenocarcinoma, non-solid type | 78 |
| Poorly differentiated adenocarcinoma, solid type | 68 |
| Well to moderately differentiated tubular adenocarcinoma | 61 |
| Signet ring cell carcinoma | 17 |
| Mucinous adenocarcinoma | 3 |

Table 1: Distribution of adenocarcinoma subtypes, which is roughly concordant with clinical distribution

## 3. Method

We trained a baseline attention-based model similar to Xu et al. (2015) which consists of a CNN encoder, acting as a feature extractor, and an RNN decoder, which produces a caption one word at a time conditioned on previous input. We defer the reader to Xu et al. (2015) for more details and implementation details. In short, for the feature extractor, we used models pre-trained on ImageNet (EfficientNetB3 (Tan and Le, 2019) and DenseNet121(Huang et al., 2017)) to extract features from the penultimate layer. We performed a 3x3 average pooling as well as a global average pooling, and then fed the extracted features into a single layer embedding encoder to reduce the dimensionality of the embedding to 256 before providing the features as input to the RNN decoder. The one-layer embedding encoder for reducing the dimensionality and the decoder were trained simultaneously. The captions were tokenised and stripped of any punctuation marks. From the extracted features, the captions were generated word by word starting from a token start word followed by sequentially updating the hidden state of the RNN model, and stopping at a token end word. The generation of each new word involved the model focusing attention on different patches or different subparts of the patches in case of the 3x3 average pooling.

We pre-extracted all the features using the pre-trained CNNs and cached them to use for training the decoder. We also extracted and cached features from patches augmented with random hue, saturation, brightness and horizontal and vertical flipping.

Figure 2 provides an overview of the inference stage. In the case of the EfficientNetB3 model, given $n$ patches from a given WSI, the output from the feature extraction for a single patch was $n \times 10 \times 10 \times 1536$. The global average pooling results in an output of $n \times 1 \times 1 \times 1536$, while the 3x3 pooling results in an output of $n \times 3 \times 3 \times 1536$. After the embedding dimensionality reduction, this becomes $n \times 1 \times 1 \times 256$ and $n \times 3 \times 3 \times 256$, respectively. These outputs are then flattened to become single dimensional vectors to be fed as input in the RNN model. This is similar for the DenseNet121 model, except the output from the CNN was $n \times 9 \times 9 \times 1024$.

We trained the model using categorical cross entropy loss with Adam optimisation algorithm (Kingma and Ba, 2014) with a learning rate of 0.001 and a decay rate of 0.99 every epoch. Each WSI had a variable number of patches, resulting in a variable input size into the RNN. We used a batch size of one. The model with the highest (bilingual evaluation understudy) BLEU@4 score (Papineni et al., 2002) on the validation set was chosen as the final model. The BLEU score is a commonly used metric for evaluating the quality of pairs of texts or text which has been machine-translated from one natural language to another. BLEU was one of the first metrics proposed and is reported to have a high correlation with human judgements of quality. The BLEU@4 is a score between 0 and 1, and it indicates how similar the candidate text is to the reference texts, with values closer to 1 representing more similar texts. Scores over 0.3 generally reflect understandable translations, and scores over 0.5 represent adequate or high quality translations. Table A3 provides guideline interpretations of the BLEU scores.

## 4. Results

Overall we ran a total of eight variations: two model architectures (DenseNet121 and EfficientNetB3), two pooling strategies (3x3 and global average pooling, referred to p3x3 and

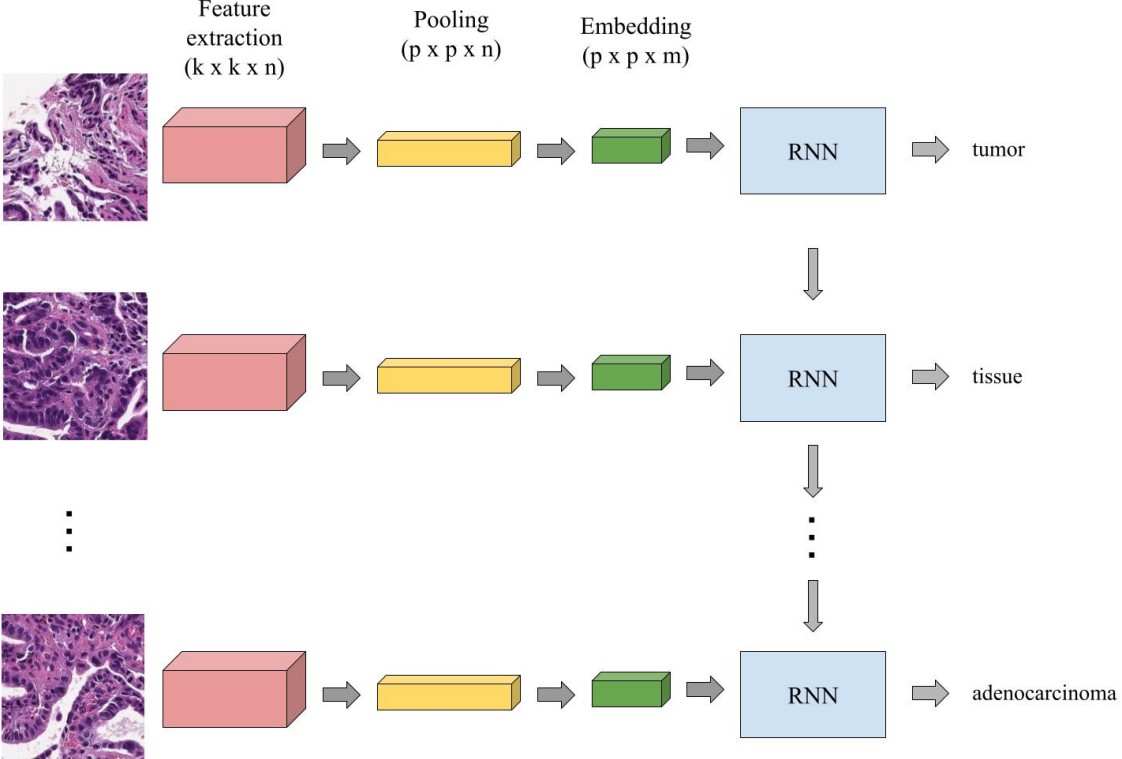

Figure 2: Given patches from a WSI, we extract features using a pre-trained CNN, which we then pool and reduce the dimensionality of. The caption is then generated by feeding the features into an RNN model step by step while updating its hidden state. In the case of the EfficientNetB3 model, given $n$ patches from a given WSI, the output from the feature extraction for a single patch was $n \times 10 \times 10 \times 1536$. The global average pooling results in an output of $n \times 1 \times 1 \times 1536$, while the 3x3 pooling results in an output of $n \times 3 \times 3 \times 1536$. After the embedding dimensionality reduction, this becomes $n \times 1 \times 1 \times 256$ and $n \times 3 \times 3 \times 256$, respectively. These outputs are then flattened to become single dimensional vectors to be fed as input in the RNN model.

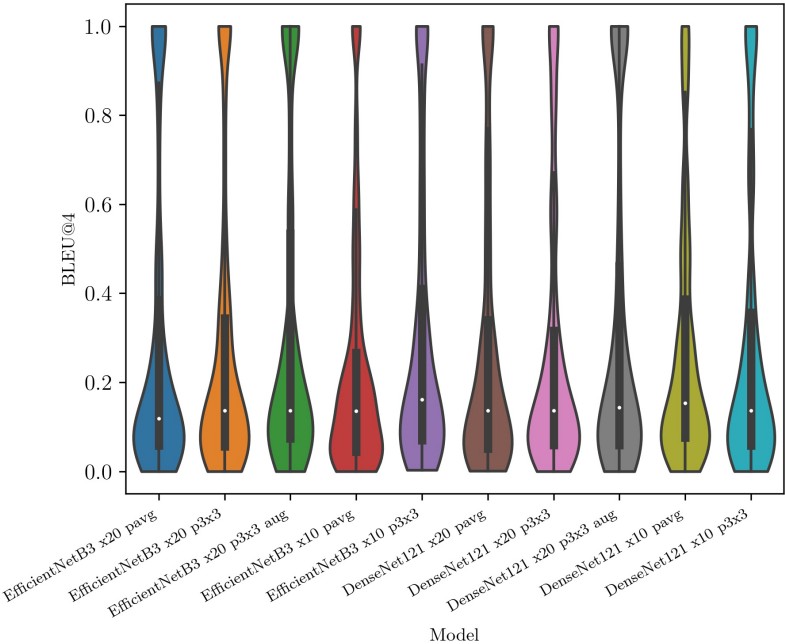

Figure 3: Evaluation of BLEU@4 on the test set consisting of 198 cases

pavg), with/without augmentation, and two magnifications (x10 and x20). We ran each experiment three times and reported the average of the three runs. We computed the BLEU@4 score between the ground truth captions and the predicted captions. We also computed a score measuring the average score of 1-gram and 2-gram (avg. 1/2-gram) word overlaps of the occurrence of the subtype class in the predicted captions; this is to measure if at least any of the shorter words that occur in the subtype name occured in the predicted caption. We evaluated the models on the test set and summarised the results in Fig. 3, Fig. 4 and Tab. 2. The best model as indicated by Tab. 2 was the EfficinetNetB3 x20 with 3x3 average pooling, achieving a BLEU@4 score of 0.324 (± 0.354) and an average 1/2 gram of 0.744(± 0.269). Table C6 provides examples of prediction outputs on selected cases from the test set.

## 5. Discussion

In this paper, we have explored the application of RNN-based attention models for the task of generating captions from a collective set of patches, rather than just a single patch. This setting corresponds to the use of existing data extracted from medical records rather than actively creating an annotated captioned dataset from scratch, which tends to be more time consuming. The occurrence of the subtype in the predicted caption is highly encouraging (see 4), with the highest score achieved (in both BLEU@4 and 1/2-gram) by the EfficientNetB3 model at x20 with augmentation and 3x3 pooling rather than global average pooling. This potentially gives the model more granularity to focus on image regions. In terms of caption prediction, there was a large spread of performance across

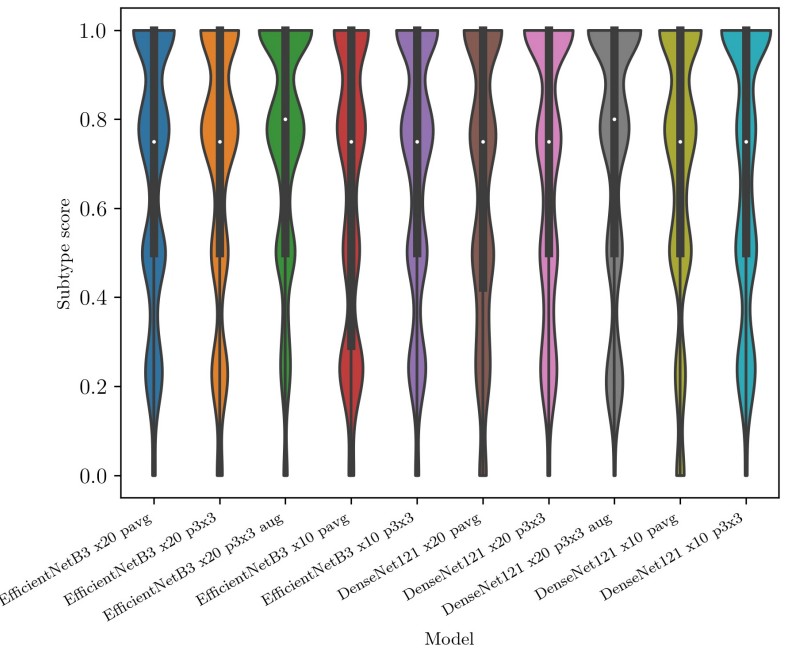

Figure 4: Avg. 1/2-gram overlaps of the occurrence of the subtype class in the predicted captions

|  | BLEU@4, mean (SD) | Avg 1/2-gram, mean (SD) |
| --- | --- | --- |
| DenseNet121 x10 p3x3 | 0.283 (0.319) | 0.695 (0.303) |
| DenseNet121 x10 pavg | 0.273 (0.281) | 0.696 (0.300) |
| DenseNet121 x20 p3x3 | 0.266 (0.300) | 0.699 (0.298) |
| DenseNet121 x20 p3x3 aug | 0.323 (0.356) | 0.729 (0.295) |
| DenseNet121 x20 pavg | 0.272 (0.311) | 0.652 (0.311) |
| EfficientNetB3 x10 p3x3 | 0.302 (0.318) | 0.689 (0.289) |
| EfficientNetB3 x10 pavg | 0.229 (0.267) | 0.643 (0.315) |
| EfficientNetB3 x20 p3x3 | 0.270 (0.313) | 0.686 (0.297) |
| EfficientNetB3 x20 p3x3 aug | **0.324 (0.354)** | **0.744 (0.269)** |
| EfficientNetB3 x20 pavg | 0.283 (0.334) | 0.682 (0.292) |

Table 2: BLEU@4 and avg. 1/2-gram scores for the different model variations.

cases. The highest average BLEU@4 was 0.324 (0.354), which, based on the guideline interpretations of BLEU, is an understandable to good match. However, the standard deviation is quite high, with some caption having a score less that 0.1, which is quite low and represents almost useless predictions. The higher average 1/2-gram indicates that the subtype occurred more frequently in the predicted caption, and that mostly the errors in prediction tended to be more in the morphological description. When grouping the scores by subtypes, the subtypes that occured more frequently in the dataset, such as well and moderately differentiated tubular adenocarcinoma, had higher BLEU@4 scores (see Tab. B4 and B5). Unsurprisingly, this indicates that a larger training dataset with well represented subtypes would lead to a better expected performance. While we did not do this, the feature extractor may benefit from further fine-tuning, potentially by using the subtype labels as targets, which could make it more sensitive to the underrepresented subtypes.

While the results obtained in this paper are encouraging, it is still far from reaching a level acceptable for use in a clinical setting. This study has a few limitations. One limitation of the dataset is that there was only a single caption per WSI, some of which with nearly identical phrasing, given that they originated from a single hospital, which limited the variety of outputs from the model. Model training typically benefit from the presence of multiple captions per WSI. This, nonetheless, highlights the potential challenging task where the goal would be to train models from existing unstructured data which originates from multiple sources without having to request from pathologists to manually re-annotate thousands of images. Another limitation is that we did not explore in this paper the use of the plethora of more recent advances in vision-language models (e.g. transformers), which could potentially lead to some improvement in performance; however, we sought to demonstrate the application of an out-of-the-box feature extractor and a baseline RNN model for generating captions on a dataset of captions that we have extracted from medical records. We have publicly released this dataset, and we hope that it will be useful for future research. While the scope of this study was limited to adenocarcinoma, we envisage that training a model on a large dataset of WSIs with diagnostic reports from different hospitals and a wide variety of cancers could lead to improved performance and make it easier to adopt such models in a clinical workflow.

## Acknowledgments

For obtaining the slides, we are grateful for the support of Takayuki Shiomi and Ichiro Mori, Department of Pathology, Faculty of Medicine, and Ryosuke Matsuoka, Diagnostic Pathology Center, at International University of Health and Welfare, Mita Hospital.

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

## Appendix A. BLEU score

| BLEU score | Interpretation |
| --- | --- |
| <0.10 | Almost useless |
| 0.10 - 0.19 | Hard to get the gist |
| 0.20 - 0.29 | The gist is clear, but has significant grammatical errors |
| 0.30 - 0.40 | Understandable to good translations |
| 0.40 - 0.50 | High quality translations |
| 0.50 - 0.60 | Very high quality, adequate, and fluent translations |
| >0.60 | Quality often better than human |

Table A3: Guideline interpretation of BLEU scores. Adapted from (goo, 2022)

## Appendix B. Dataset

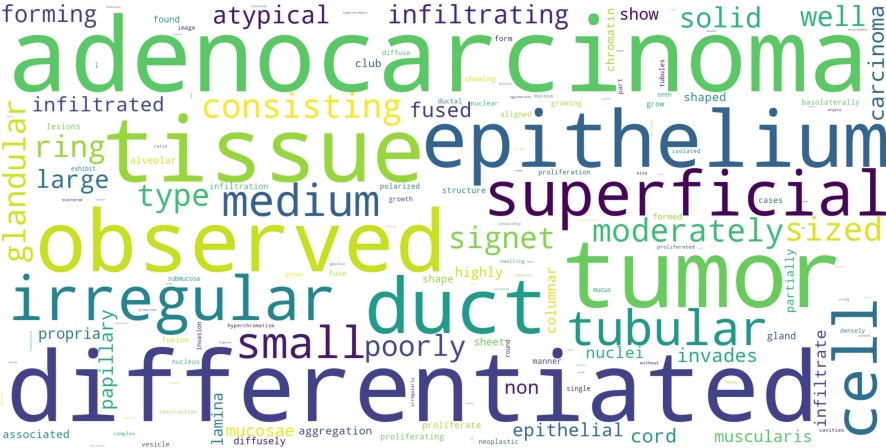

Figure B5: Word cloud of commonly occurring words

Table B4: Well differentiated tubular adenocarcinoma

|  | BLEU@4, mean (SD) | Subtype ngram@2, mean (SD) |
| --- | --- | --- |
| DenseNet121 x10 p3x3 | 0.344 (0.345) | 0.798 (0.308) |
| DenseNet121 x10 pavg | 0.297 (0.287) | 0.754 (0.289) |
| DenseNet121 x20 p3x3 | 0.344 (0.343) | 0.825 (0.279) |
| DenseNet121 x20 p3x3 aug | 0.474 (0.402) | 0.846 (0.262) |
| DenseNet121 x20 pavg | 0.354 (0.343) | 0.772 (0.325) |
| EfficientNetB3 x10 p3x3 | 0.383 (0.375) | 0.741 (0.327) |
| EfficientNetB3 x10 pavg | 0.185 (0.226) | 0.575 (0.353) |
| EfficientNetB3 x20 p3x3 | 0.439 (0.400) | 0.842 (0.278) |
| EfficientNetB3 x20 p3x3 aug | 0.436 (0.401) | 0.816 (0.269) |
| EfficientNetB3 x20 pavg | 0.405 (0.393) | 0.746 (0.301) |

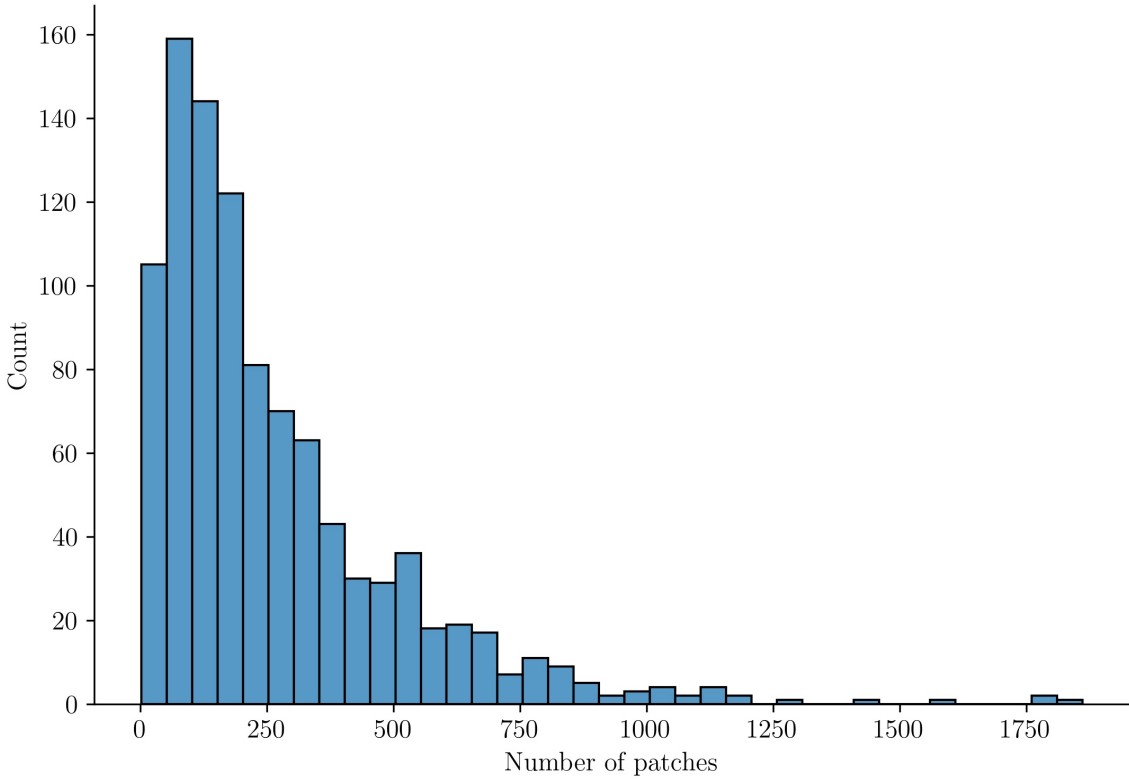

Figure B6: Counts of patches per WSI. Each WSI has a single caption.

Table B5: Moderately differentiated tubular adenocarcinoma

| | BLEU@4, mean (SD) | Subtype ngram@2, mean (SD) |
|---|---|---|
| DenseNet121 x10 p3x3 | 0.347 (0.353) | 0.759 (0.294) |
| DenseNet121 x10 pavg | 0.374 (0.357) | 0.759 (0.336) |
| DenseNet121 x20 p3x3 | 0.367 (0.323) | 0.783 (0.255) |
| DenseNet121 x20 p3x3 aug | 0.388 (0.368) | 0.849 (0.204) |
| DenseNet121 x20 pavg | 0.408 (0.367) | 0.750 (0.282) |
| EfficientNetB3 x10 p3x3 | 0.399 (0.329) | 0.717 (0.286) |
| EfficientNetB3 x10 pavg | 0.333 (0.334) | 0.750 (0.310) |
| EfficientNetB3 x20 p3x3 | 0.259 (0.320) | 0.675 (0.280) |
| EfficientNetB3 x20 p3x3 aug | 0.375 (0.379) | 0.811 (0.281) |
| EfficientNetB3 x20 pavg | 0.392 (0.388) | 0.783 (0.290) |

## Appendix C. Example prediction results

| BLEU@4 | Image id | Original caption | Predicted caption |
|---|---|---|---|
| 0 | 026a5c88 b93b4669 b2c8d793 32feab7b | atypical l cells with atypical nuclei forming diffuse or irregular atypical ducts and infiltrating poorly differentiated adenocarcinoma solid type | in the superficial epithelium in the superficial epithelium in the superficial epithelium in the superficial epithelium in the superficial epithelium in the superficial epithelium in the superficial epithelium in the superficial epithelium in the superficial epithelium in the superficial epithelium in the superficial epithelium |
| 0.1 | e9b41535 519f4f60 848171ba 0a4fcabe | tumor tissue consisting of medium sized cord like or irregular glandular infiltration is observed in the superficial epithelium poorly differentiated adenocarcinoma non solid type or moderately differentiated tubular adenocarcinoma | medium to small irregular ducts is observed moderately differentiated tubular adenocarcinoma |
| 0.3 | d36d795d d5ab4833 b0022ccd d025abc3 | on the superficial epithelium tumor tissue that infiltrates by forming medium sized papillary or small irregular ducts is observed papillary adenocarcinoma | on the superficial epithelium tumor tissue consisting of medium sized irregular or large and small ducts papillary adenocarcinoma |
| 0.56 | 1c260429 f2ce4ee3 96093d43 26293315 | from the superficial epithelium to the muscularis mucosae tumor tissue consisting of medium sized and irregular glandular ducts infiltrating is observed well differentiated tubular adenocarcinoma | from the superficial epithelium to the muscularis mucosae tumor tissue consisting of medium sized and irregular invades by forming medium sized to small irregular ducts is observed moderately differentiated tubular adenocarcinoma |
| 0.92 | 7c6008a2 7bb3452b b04045bd 5f96d864 | in the superficial epithelium tumor tissue that invades by forming medium sized to small irregular ducts is observed moderately differentiated adenocarcinoma | in the superficial epithelium tumor tissue that invades by forming medium sized to small irregular ducts is observed moderately differentiated tubular adenocarcinoma |
| 1 | df1656e6 20c44e15 abbfdebc bcdc8e6b | tumor tissue consisting of cord like or small irregular glandular ducts fused and infiltrated is observed in the superficial epithelium poorly differentiated adenocarcinoma non solid type | tumor tissue consisting of cord like or small irregular glandular ducts fused and infiltrated is observed in the superficial epithelium poorly differentiated adenocarcinoma non solid type |

Table C6: Example of caption prediction results ranging from 0 to 1 in the BLEU@4 score on six cases used in the test set.

