# OpenReview forum: "Inference of captions from histopathological patches"
_MIDL.io/2022/Conference — MIDL 2022_

### Official Review · Reviewer_6nhT · 2022-01-05

**Confidence:** 3
**Preliminary Rating:** 2
**Recommendation:** Poster

**Summary:**

The paper describes a rather large (7 GB) dataset of gastric adenocarcinoma patches cropped from WSIs alongside with a pathology report and a tumor subtype category. The authors further describe a baseline experiment, using of a pre-trained CNN encoder, a translation layer and a RNN-based report/token generator. The task itself (pathology report generation) is very interesting and a plethora of training data should be there to harvest in clinical archives around the world.

**Strengths:**

The paper is written well and clearly states the experiments that were conducted. The dataset itself has a nice size and might be of use also for other tasks. The baseline method is sound for this kind of task. The paper features a nice discussion.

**Weaknesses:**

- The paper aims at publishing a dataset and provides a baseline experiment for it. Yet, for a dataset publication, there are some weaknesses that I would like to highlight. The patches were extracted using a not clearly described methodology from WSIs, yet the WSIs were not provided. This means that spatial information within the WSI is no longer available. The size of the WSIs should not be orders of magnitudes higher, so I would suggest to publish the WSI themselves, so people are more free as to their algorithmic choices. It also resembles the real use case.

- I am not an expert in NLP, but from my understanding (and also from what the authors state in their discussion) the results are rather poor, and from the paper it is not clear why this is the case. The paper does neither provide example results nor provide insights on why the results were as they were.

- The authors cite related work yet miss to report any baseline results besides their own toolchain. This would be good to evaluate the dataset quality.

- The authors did not provide any baseline results of their method on other datasets (e.g., TCGA pathology reports)

- Opposed to what the authors state, the code is not available on their github repository.



**Deanonymize Review:**

no

**Detailed Comments:**

- I am not sure if the BLEU-score is the most proper evaluation for a task like this, as there might be many „correct“ ways of describing the findings in the pathology report, yet different readers will describe with varying levels of details or even just use very different terms (with the same meaning to a human reader).  It is in this context a pity that the authors did not provide any examples of their results.

- If you need some space in the paper: Your list of citations on the first page is rather extensive and could be shortened.

**Final Rating After The Rebuttal:**

4: Weak Accept

**Justification Of The Final Rating:**

The authors addressed most of the comments in their revised version. I still think that it's a pity to only have patches, as this drastically limits the use of the dataset for approaches that take into account a wider field of view (like, from my experience, pathologists will do)

**Paper Type:**

validation/application paper

**Questions To Address In The Rebuttal:**

- Please provide an argumentation on why it was not possible to publish the WSIs and the patch extraction method but just the patches.

- Please provide an argumentation on why the dataset itself is of value, given the rather poor baseline results of your method.

- Please indicate why you did not conduct experiments either trialing your method on other datasets (e.g., pathology reports from TCGA) or evaluate your dataset using baseline methods from other authors.

- Please make available your code as well (as you wrote in your paper).



**Special Issue:**

no

---

### Official Review · Reviewer_gSfN · 2022-01-11

**Confidence:** 4
**Preliminary Rating:** 4
**Recommendation:** Poster

**Summary:**

This paper explores the automatic generation of diagnostic reports from histopathological images. The authors propose a new publicly available dataset with 991 WSIs with captions extracted from the reports, and split into a total of 262K patches. No novelty in the method, but well designed baseline and good evaluation.


**Strengths:**

- The authors propose a new publicly-available and well-prepared dataset (it would be even better to also share the code publicly)
- A simple but well-designed baseline is proposed
- Good evaluation of the models
- Overall well written


**Weaknesses:**

- There is no technical novelty
- The paper lacks some descriptions and discussion (see detailed comments below)
- The dataset originated from a single hospital, resulting in a low variation in the images and reports



**Deanonymize Review:**

no

**Detailed Comments:**

- Maybe mention diagnostic reports in the title, as it is more specific than captions.
- Briefly introduce the metrics, i.e. their range and what they evaluate.
- 260K patches in the abstract, then 262K in the introduction
- I think saying that you release 260K patches is misleading. You have 991 WSIs
- In 2., mention how many experts produced the reports.
- In 3. It is not clear that you apply and compare two different pooling strategies. It reads as it could be one after the other. It becomes clearer only later with the results. Maybe specify that p=1 in Fig. 2 in the case of global average pooling, and its value in the case of local 3x3 pooling.
- In 3. “pre-trained models on ImageNet”-> models pre-trained on ImageNet.
- Report best results in bold in the tables
- There is no discussion of the 1/2-gram results in 5.
- Some typos in the discussion, e.g. “This settings”, “more recent the more recent”, “within cases” should be across cases.


**Paper Type:**

validation/application paper

**Questions To Address In The Rebuttal:**

All comments above, although some minor, should be addressed to improve the quality. Except for the lack of technical novelty and origin from a single hospital, which I do not think the authors can address in the rebuttal but could be acceptable limitations of the paper.


**Special Issue:**

no

---

### Official Review · Reviewer_V1CR · 2022-01-23

**Confidence:** 3
**Preliminary Rating:** 4
**Recommendation:** Poster

**Summary:**

The authors present a image captioning dataset in the context of histopathology. The dataset consists of retrospectively collected images from adenocarcinoma biopsy specimen with a of 260k patches from 991 WSI slides (H&E stained, from distinct patients) with associated diagnostic captions. The authors further describe a baseline for the captioning task using a simple attention-based CNN-RNN model.

**Strengths:**

With their paper, the authors describe both a new dataset including both imaging data and captioning as well as a corresponding baseline for image captioning, both of which they make publicly available (although the code is not yet available on github). The description of their dataset is straightforward and well structured, the results include a simple but extensive baseline.

**Weaknesses:**

- The authors split the WSIs in patches of 300 x 300 px, and assigned labels according to the WSI label. It was not clear to the reviewer why this patch size was chosen. A discussion is missing on whether this field of view is generally sufficient to generate the slide-level caption. One alternative could be to provide the full WSI images + a mask that delineates the cancer regions (which the authors have used to extract the patches from adenocaricoma regions).
- The image patches are currently linked via a google drive folder - potentially, an image data base / image repository (e.g., TGCA, Zenodo, figshare) would be a good alternative to host the data set continuously, including potential versioning of the dataset. A short description of the data repository, e.g., in the supplementary material would be helpful to have the most important information in one spot.
- While the purpose of the baseline is clear, transformer architectures have mostly become the state of the art in the field of image captioning and implementations are publicly available. Here, a more "recent" baseline would have been nice to see.


**Deanonymize Review:**

yes

**Detailed Comments:**

Additional suggestions:
- For future referencing, giving the data set a "name" makes it easier to recognize and reuse.
- it seems as if the ARCH data set (described in the paper by Gamper and Rajpoot) seems to be a closely related work, and additional comments on the similarity and the differences (of which their are plenty) between the two datasets would strengthen the paper-
- Additional details on the data generation could be added, e.g., who translated the reports to English? Who outlined the cancer regions / what were the criteria for that? Why did some WSIs contain very few cancer patches (50-100)?
- What meta data is available for the slides? Is the subtype always available in the caption?
- To the reviewer, the training strategy was not fully clear: On the one hand, the authors state "The embedding encoder and decoder were trained simultaneously.", then "We pre-extracted all the features using the pre-trained CNNs and cached them to use for training the decoder." - This could be phrased more clearly. Furthermore, the figure seems to indicate that for each new token, a new image patch is used to generate the next token. How is this implemented during training and inference? How is it decided how many patches are presented to the network? Is a stopping token defined? This could be explained more clearly.
- A short textual summary of the results in section 4 is missing and should highlight the most important observations.
- As the BLEU score may not be as widely known in the medical imaging community, a short explanation of why it is applicable, and a short discussion of the values achieved would be highly appreciated (partially present in the discussion). Additional examples with generated text / ground truth would be interesting.

Minor comments, typos, etc.:
- "to Xu et al. (2015)’s" - why 's ?
- "This settings" - typo
- "which tends to be more time consuming; however, it is more likely to lead to better performance" - this is a bit unclear and could be phrased more straightforwardly.

**Final Rating After The Rebuttal:**

4: Weak Accept

**Justification Of The Final Rating:**

The authors addressed most comments in their rebuttal and clarified open points. I am happy to see the dataset now on Zenodo. Still, I agree with Reviewer 6nhT that the use of patches will limit the flexibility of use of the dataset drastically. I would support the change of title suggested by Reviewer gSfN.


**Paper Type:**

validation/application paper

**Questions To Address In The Rebuttal:**

The authors present an interesting dataset, but the main point of criticism would be the decision for 300x300px image patches, which should be explained / justified or potentially abandoned in favour of uploading the WSIs. Furthermore, the authors may want to comment on providing the dataset on an hosting platform which also allows straightforward versioning of the dataset.

**Special Issue:**

no

---

### Meta-Review · Area_Chair_M1vt · 2022-02-15

**Recommendation:** Accept (Poster)
**Confidence:** 5

**Metareview:**

The authors' have provided significant improvement to their original work by addressing all the comments raised by the reviewers. This has also resulted in a unanimous agreement among the reviewers for the acceptance of this work to the 2022 MIDL conference as a poster presentation. The image patches are also provided as open-source which is encouraging. I do not have any objections to changing the title as it was deemed more suitable for the proposed work.

---

### Decision · Program_Chairs · 2022-02-28

Accept